

# Declines in skeletal muscle quality vs. size following two weeks of knee joint immobilization

Rob J. MacLennan[1], Michael Sahebi[1], Nathan Becker[1], Ethan Davis[1], Jeanette M. Garcia[2] and Matt S. Stock[1]

[1] School of Kinesiology and Physical Therapy, University of Central Florida, Orlando, FL, United States of America
[2] Department of Health Sciences, University of Central Florida, Orlando, FL, United States of America

## ABSTRACT

**Background.** Disuse of a muscle group, which occurs during bedrest, spaceflight, and limb immobilization, results in atrophy. It is unclear, however, if the magnitude of decline in skeletal muscle quality is similar to that for muscle size. The purpose of this study was to examine the effects of two weeks of knee joint immobilization on vastus lateralis and rectus femoris echo intensity and cross-sectional area.

**Methods.** Thirteen females (mean ± SD age = 21 ± 2 years) underwent two weeks of left knee joint immobilization via ambulating on crutches and use of a brace. B-mode ultrasonography was utilized to obtain transverse plane images of the immobilized and control vastus lateralis and rectus femoris at pretest and following immobilization. Effect size statistics and two-way repeated measures analyses of variance were used to interpret the data.

**Results.** No meaningful changes were demonstrated for the control limb and the rectus femoris of the immobilized limb. Analyses showed a large increase in vastus lateralis echo intensity (i.e., decreased muscle quality) for the immobilized limb ($p = .006$, Cohen's $d = .918$). For vastus lateralis cross-sectional area, no time × limb interaction was observed ($p = .103$), but the effect size was moderate ($d = .570$). There was a significant association between the increase in vastus lateralis echo intensity and the decrease in cross-sectional area ($r = -.649$, $p = .016$).

**Conclusion.** In female participants, two weeks of knee joint immobilization resulted in greater deterioration of muscle quality than muscle size. Echo intensity appears to be an attractive clinical tool for monitoring muscle quality during disuse.

Corresponding author
Matt S. Stock, matt.stock@ucf.edu

## INTRODUCTION

Muscle unloading or disuse, such as what occurs during bed rest and removal of weight bearing conditions, results in muscular atrophy (*Berg, Larsson & Tesch, 1997*; *Tesch, Trieschmann & Ekberg, 2004*; *Wall et al., 2014*). These changes present several significant challenges for health care providers, particularly following surgical procedures and bedrest in older adults, as muscle weakness is a strong predictor of mortality (*Newman et al., 2006*). Responses to disuse vary by sex (*Deschenes et al., 2009*) and muscle group (*Schulze,*

*Gallagher & Trappe, 2002*; *Tesch, Trieschmann & Ekberg, 2004*), with greater declines in females and the knee joint, respectively. In athletics, females have a roughly three times greater incidence of anterior cruciate ligament (ACL) tears during soccer and basketball versus males, with many of these injuries occurring during non-contact situations (*Prodromos et al., 2007*; *Renstrom et al., 2008*). Female athletes that play sports year-round have an ACL tear rate of approximately 5% (*Renstrom et al., 2008*). These issues are not limited to competitive athletes, however. ACL tears are common among any population that engages in physical activity and sports. It has been reported that the overall age- and sex-adjusted annual incidence of ACL tears among the general population was 68.6 per 100,000 person-years, with a substantial increase over time regardless of age (*Sanders et al., 2016*). Rehabilitation of the knee joint following surgical interventions has major social and economic consequences, with one study estimating a cost of >$2 billion for an annual incidence of 200,000 ACL reconstructions in the U.S (*Mather et al., 2013*). Following ACL reconstruction surgery, specifically with a concomitant meniscal repair, patients are often placed under temporary bed rest to maintain graft integrity, as well as reduce compressive forces and pain at the knee joint (*Hiemstra et al., 2009*). During short-term immobilization, patients are typically administered a knee brace to be worn for all weight bearing activities.

Given the implications for long-term health, changes in muscle size (i.e., hypertrophy and atrophy) following various interventions have historically garnered substantial interest from researchers and clinicians. In recent years, however, investigators have begun to consider changes in muscle quality via analysis of echo intensity (*Lopez et al., 2017*). The premise behind its use is that the degree of darkness/brightness of a given region of interest characterizes tissue composition, as darker images (lower values) represent skeletal muscle tissue and brighter images (higher values) represent non-contractile elements. While echo intensity's precise underpinnings have yet to be directly verified, anatomical investigations carried out in humans and animals have demonstrated that it strongly reflects fibrous tissue content (*Arts et al., 2012*; *Pillen et al., 2009*), as well as intramuscular adiposity (*Nishihara et al., 2014*; *Reimers et al., 1993*; *Young et al., 2015*). Echo intensity also seems to be at least moderately correlated with performance measures such as muscle strength (*Mota, Stock & Thompson, 2017*), stair-climbing ability (*Kleinberg et al., 2016*), and chair stand ability (*Rech et al., 2014*; *Lopez et al., 2017*), and these associations have been reported from adolescents (*Stock et al., 2017*) through older adults (*Watanabe et al., 2013*). Furthermore, echo intensity can be easily analyzed with B-mode ultrasonography. Though its image quality still lags behind CT and MRI, ultrasonography's portability, cost, and time efficiency make it a realistic option for clinicians and practitioners with limited resources. While more research and normative data are needed, at this point, ultrasonography-based echo intensity shows promise as a tool for tracking changes in muscle quality.

Understanding the neuromuscular and morphological adaptations that occur during short-term disuse will allow for the development of more effective and timely preventative interventions. While there is evidence that resistance training may affect echo intensity in older people (*Radaelli et al., 2013*), evidence of this relationship in young people is less clear. Furthermore, as additional imaging analytical techniques become available, it

is important to understand which variables may be more or less sensitive to changes in muscle morphology, particularly in clinical settings. While the atrophic effects of limb immobilization have been well established (*Hackney & Ploutz-Snyder, 2012*), the extent to which muscle quality deteriorates is unclear. Thus, the purpose of this study was to examine the effects of two weeks of knee joint immobilization on vastus lateralis and rectus femoris muscle quality and size in college-aged females. Based on previous studies (*Deschenes, McCoy & Mangis, 2017*; *Hackney & Ploutz-Snyder, 2012*), we hypothesized that skeletal muscle size would show a small-to-moderate decrease following immobilization. In contrast, we speculated that muscle quality would deteriorate to a greater extent, which would be suggestive of rapid intramuscular fat and/or fibrous tissue infiltration during disuse, rather than reduction in muscle size.

## MATERIALS & METHODS

### Experimental approach to the problem

This investigation utilized a within-participants design in which B-mode ultrasonography imaging was performed for the vastus lateralis and rectus femoris muscles of immobilized (left) and control (right) limbs. All participants were right leg dominant. Data was collected during baseline testing (PRE) and two weeks later (POST). All visits to the laboratory occurred at the same time of day (±one hour) throughout the course of the study. All imaging and image analyses were performed by the same investigators utilizing methods described in our previous publications (*Burton & Stock, 2018*; *Mota, Stock & Thompson, 2017*). Throughout the study, participants were asked to refrain from alcohol, keep their dietary habits consistent, and comply with the immobilization protocol. Participants kept a three-day food log, which was primarily meant to serve as a reminder for the need for consistency of health behaviors throughout the study. Each participant was assigned to a study investigator, who served as her direct point of contact and was always available.

### Participants

Due to the novelty of this topic, the ability to predict the observed effect size was limited. As such, an *a priori* power analysis was not carried out as recommended by previous investigators (*Beck, 2013*). It should be noted, however, that a recent systematic review found that lower limb immobilization studies have reported data for ≤ 13 participants, with ≤ 10 being typical (*Campbell et al., 2019*). Thus, the final sample size described below is ≥ what has previously been utilized in the relevant literature. This study's *post-hoc* power analyses for particularly relevant outcomes have been included in our Supplemental Files.

Twenty-six healthy, college-aged females originally enrolled in this study. Recruitment methods included the use of flyers, social media posts, marketing on laboratory and university websites, and presentations in group settings. Inclusion criteria included females between the ages of 18–35 years, a body mass index ≤ 30 kg/m², and willingness to comply with the study's demands. Major exclusion criteria included the following nine conditions: (1) personal or family history of blood clots; (2) neuromuscular or metabolic disease; (3) osteoarthritis; (4) previous surgery on the hip or knee joints; (5) use of an assistive walking device within the previous year; (6) myocardial infarction within the past

year; (7) pregnancy; (8) use of contraceptives within the previous 90 days (*Vinogradova, Coupland & Hippisley-Cox, 2015*); (9) musculoskeletal pain or discomfort in any of the major joints. Recent lower-body resistance or aerobic training participation was not considered exclusionary; trained and untrained individuals experience similar relative declines in lean mass following knee joint immobilization (*Deschenes, McCoy & Mangis, 2017*). Participants completed an in-house Pre-Testing Health Questionnaire and the Physical Activity Readiness Questionnaire + (PAR-Q +) in the laboratory to determine eligibility. All participants read, understood, and signed an informed consent form. The study protocol was approved by the University of Central Florida's Institutional Review Board (Study # BIO-17-13642). Participants were compensated $350 for completing the study.

Out of the 26 participants that enrolled, eleven withdrew. Reasons for withdrawing included, but were not limited to, time constraints, changes in contraceptive use, difficulty accomplishing daily activities and subsequent inability to comply with the study protocol, and discomfort. Out of the fifteen participants that completed the study, two were removed from this paper's statistical analysis, one because of difficulty with image analysis and another because of accelerometer data demonstrating non-compliance (details described below). Therefore, 13 participants (mean ± SD age = 21 ± 2 years, mass = 61.6 ± 4.6 kg, height = 164.7 ± 6.1 cm) were included in our final data set.

## Immobilization procedures

During the pretesting data collection session, participants were properly fit for the use of a knee joint immobilization brace (T Scope® Premier Post-Op Knee Brace, Breg, Inc., Carlsbad, CA, USA). Each brace was locked at a 90° angle, which allowed the left knee extensors to remain relaxed and unloaded while preventing the toes from contacting the ground. A 90° knee joint angle was a conservative approach to ensuring that the toes of the left foot did not come into contact with the ground while ambulating on the crutches. Participants wore the brace at all times, except during periods of sleep. Participants were also asked to keep their left leg in the brace and covered with a plastic bag when showering. To ensure safety and comfort while showering, each participant was offered a shower chair (Medline Shower Chair Bath Seat with Padded Armrests and Back, Medline Industries, Inc., Northfield, IL, USA). Participants were also provided and properly fit for axillary crutches (Cardinal Health Axillary Crutch, Adult, Height 62–70 in, Adjustable, Cardinal Health, Inc., Dublin, OH, USA) using guidelines described by *Fairchild (2012)*. Training was provided on proper use of crutches in navigating the community including curbs, stairs, and sidewalks (Fig. 1). Participants were advised that they would likely have trouble with activities of daily living such as transportation and cleaning, to allow additional time to complete tasks, and to plan ahead in order to avoid unsafe conditions (e.g., ensure a towel was near the shower and to sufficiently dry off). In addition to use of the brace and crutches, participants were urged to refrain from weight bearing.

Each participant wore a stocking (Rolyan Extra Soft Stockinette, 100% Cotton, Performance Health, Warrenville, IL, USA) underneath their brace that spanned the length of the left lower extremity from the proximal thigh to the ankle. This stocking

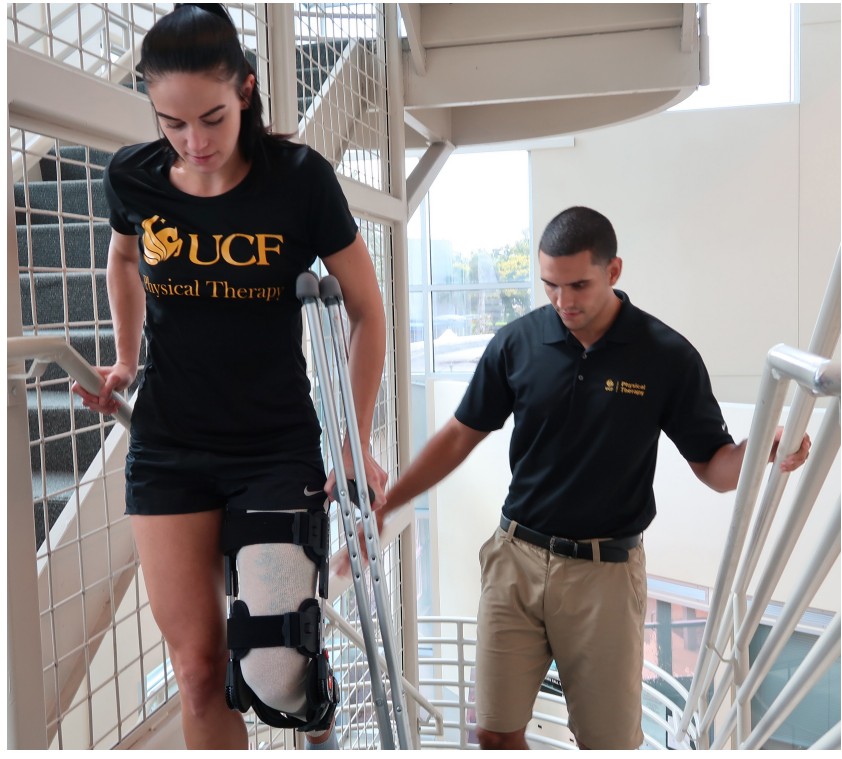

**Figure 1  Use of crutches during knee joint immoblization.** An example of a participant being instructed on proper use of crutches in navigating the community.

was worn to improve comfort while wearing the knee brace and to minimize the risk of adverse skin reactions. The stocking could be taken off before bed but was worn during all daytime activities. To reduce the risk of blood clots, participants were also given nighttime compression stockings (Medi-Pak Anti-Embolism Stockings, McKesson, San Francisco, CA, USA) to be worn over the left leg while they slept. Basic hygiene was discussed with the participants, and they were instructed to notify the researchers immediately in the event that they noticed any irritation, redness, or swelling; this was not reported by any of the participants.

Consistent with previous knee joint immobilization studies (*Deschenes, Holdren & McCoy, 2008*; *Deschenes et al., 2009*), each participant performed light range of motion movements at the ankle and knee while lying supine in bed. These activities were performed twice daily (morning and evening) in an effort to minimize the risk of vascular pathology and muscular contractures. A video containing instructions was provided to participants, as was as a handout with detailed directions of performance of the exercises. The compressive stocking was worn during the exercise. Finally, to further ensure compliance and safety, study investigators text messaged or spoke with the participants via telephone daily.

## Ultrasonography measurements and analysis

Ultrasonography images for the vastus lateralis and rectus femoris muscles of both limbs were taken with a portable B-mode imaging device (GE Logiq e BT12, GE Healthcare,

Milwaukee, WI, USA) and a multi-frequency linear-array probe (12 L-RS, 5–13 MHz, 38.4-mm field of view; GE Healthcare, Milwaukee, WI, USA). The panoramic function (LogiqView, GE Healthcare, Milwaukee, WI, USA) was used to obtain images of the vastus lateralis and rectus femoris in the transverse plane. Prior to ultrasound measurements, subjects laid in the supine position for 5 min in order to allow the redistribution of fluids from their quadriceps muscles. The images were taken at 50% of the length of the femur. A high-density foam pad was secured around the thigh with an adjustable strap to ensure probe movement in the transverse plane. Ultrasonography settings (Frequency: 10 MHz, Gain: 55 dB, Dynamic Range: 72) were kept consistent across participants. To ensure optimal image clarity, a standardized depth of 5.0 cm was utilized. However, for three participants, a greater depth was necessary to adequately capture each muscle belly. For these three participants, depths of 6.0, 6.0, and 7.0 cm were utilized. Depth for each participant was kept consistent across trials. A generous amount of water-soluble transmission gel (Aquasonic 100 ultrasonography transmission gel, Parker Laboratories, Inc., Fairfield, NJ, USA) was applied to the skin to allow immersion of the probe surface during measurement to enhance acoustic coupling. Three images of each muscle were obtained, and mean values were used for statistical analyses.

The ultrasonography images were digitized and examined with ImageJ software (version 1.46, National Institutes of Health, Bethesda, MD, USA) at the conclusion of the study. The polygon function was used to outline the borders of the vastus lateralis and rectus femoris. After scaling the image units from pixels to cm, muscle cross-sectional area ($cm^2$) was determined with the polygon function. Echo intensity was also assessed by computer-aided gray-scale analysis using the histogram function. The echo intensity values were determined as the corresponding index of muscle quality ranging between 0 and 255 arbitrary units (AU). Recently, *Young et al. (2015)* provided an equation that allows investigators to correct rectus femoris echo intensity for the magnitude of subcutaneous thickness over the muscle (corrected echo intensity = raw echo intensity + [subcutaneous fat thickness [cm] × 40.5278]). While uncorrected rectus femoris echo intensity values have been reported herein, it should be noted that correction for subcutaneous thickness would not have affected this study's overall conclusions.

## Ultrasonography test-retest reliability

Test-retest reliability statistics were calculated for each of the dependent variables using the equations described by *Weir (2005)*. Specifically, we assessed the intraclass correlation coefficient (ICC [model 2, k]), standard error of measurement (SEM [expressed both in absolute terms and as a percentage of the mean]), and the minimal difference needed to be considered real (MD). As presented in a recent publication (*Burton & Stock, 2018*), our laboratory's test-retest reliability statistics for rectus femoris echo intensity in college-aged females were: ICC = .928; SEM = 4.21 AU (4.22%); MD = 11.68 AU. For rectus femoris cross-sectional area, these statistics were as follows (*Burton & Stock, 2018*): ICC = .982; SEM = 0.24 $cm^2$ (5.43%); MD = 0.67 $cm^2$. For the vastus lateralis, the echo intensity statistics were: ICC = .810; SEM = 3.31 (3.71%); MD = 9.17 AU. The test-retest reliability

statistics for vastus lateralis cross-sectional area were: ICC = .980; SEM = 0.52 (3.43%); MD = 1.45 cm$^2$.

## Actigraphy

In order to assess compliance with the established protocol, participants were asked to wear Actigraph GT9X accelerometers (ActiGraph Inc, Penscola, FL, USA) on both their right and left legs, fastened securely around each ankle via Velcro straps. The accelerometers detect normal human motion while filtering out high-frequency vibrations that would artificially increase movement data. Participants were instructed to wear accelerometers around both ankles for the entire study period, only removing the devices during showering. Two levels of compliance were established for the current protocol: (1) overall wear-time compliance and (2) compliance to the immobilization protocol. To meet wear-time compliance criteria, established by *Troiano (2007)*, participants were required to wear the device for a minimum of four days (10 h per day) over a 7-day period. As participants were instructed to wear the device over a two-week period, eight full days of data, including two weekend days, were required to meet compliance criteria. To determine compliance to the immobilization protocol, criteria was established based on a previous study by *Cook, Clark & Ploutz-Snyder (2006)*, who examined differences in both the number of steps and intensity of steps between legs in non-weight bearing participants on crutches.

## Statistical analyses

All variables were first checked for normal distributions with Shapiro–Wilk tests. To assess compliance with protocol, paired samples t-tests were conducted to determine differences in activity intensity and steps between the immobilized and control limbs of each participant. Two-way (time [pre, post] × limb [immobilized, control]) repeated measures analyses of variance (ANOVAs) were used to examine mean differences for each muscle and dependent variable. When appropriate, follow-up analyses included Bonferroni post-hoc comparisons and paired samples *t*-tests. We also evaluated 95% confidence intervals for mean differences (CIs) and effect sizes via Cohen's *d* statistics. Small, medium, and large Cohen's d corresponded to values of 0.20, 0.50, and 0.80, respectively (*Cohen, 1988*). All statistical procedures were carried out with JASP software (version 0.8.3.1, University of Amsterdam, Amsterdam, The Netherlands).

# RESULTS

Mean ± SD values for all ultrasonography variables have been displayed in Table 1.

## Vastus lateralis echo intensity

The results from the two-way repeated measures ANOVA indicated that there was a limb × time interaction ($F = 9.14$, $p = .011$ (Fig. 2)). Follow-up analyses revealed a significant increase in echo intensity ($p = .006$, 95% CI [−5.60 to −1.16 AU]) for the immobilized limb that was considered large (mean increase = 3.40 AU, $d = .918$). In contrast, there was a small decrease in echo intensity for the control limb that was not significant (mean decrease = 1.87 AU, $p = .189$, $d = .386$, 95% CI [−1.06–4.79 AU]). Differences between

**Table 1  Two weeks of knee joint immoblization result in significant increases in vastus lateralis echo intensity.** Mean ± SD echo intensity and cross-sectional area values for each muscle and testing session of the immoblized (A.) and control (B.) limbs. * = $p < .05$.

### A. Immobilized Limb

| Vastus lateralis cross-sectional area (cm²) | | Rectus femoris cross-sectional area (cm²) | |
| --- | --- | --- | --- |
| PRE | POST | PRE | POST |
| 16.71 ± 2.85 | 15.60 ± 2.38 | 6.15 ± 1.17 | 5.93 ± 0.86 |
| Vastus lateralis echo intensity (AU) | | Rectus femoris echo intensity (AU) | |
| PRE | POST | PRE | POST |
| 82.77 ± 8.35 | 86.17 ± 9.54* | 82.85 ± 7.61 | 83.25 ± 10.30 |

### B. Control Limb

| Vastus lateralis cross-sectional area (cm²) | | Rectus femoris cross-sectional area (cm²) | |
| --- | --- | --- | --- |
| PRE | POST | PRE | POST |
| 17.05 ± 3.09 | 16.84 ± 3.59 | 6.17 ± 1.44 | 5.95 ± 1.42 |
| Vastus lateralis echo intensity (AU) | | Rectus femoris echo intensity (AU) | |
| PRE | POST | PRE | POST |
| 86.13 ± 10.47 | 84.26 ± 9.25 | 83.95 ± 10.90 | 83.38 ± 12.21 |

limbs were small/moderate but not significant at the pretest ($p = .064$, $d = .567$, 95% CI [−0.22–6.94 AU]) or posttest ($p = .169$, $d = .406$, 95% CI [−0.93–4.74 AU]). For the immobilized limb, three participants showed an increase in vastus lateralis echo intensity that exceeded the MD (9.17 AU). The mean increase in echo intensity for the immobilized limb (3.40 AU) did not exceed the MD.

## Vastus lateralis cross-sectional area

Despite a large effect size, the results from the two-way repeated measures ANOVA indicated that there was no limb × time interaction ($F = 3.11$, $p = .103$ (Fig. 3)) and no main effects for limb ($F = 1.81$, $p = .204$) or time ($F = 2.75$, $p = .123$). Analysis of effect sizes and 95% CIs for mean differences revealed similar cross-sectional area values for the pretest and posttest for the control limb ($d = .151$, 95% CI [−0.64 −1.08 cm²]). However, a moderate effect size was found for the immobilized limb ($d = .570$, 95% CI [−0.07–2.28 cm²]). For the immobilized limb, five participants showed a decrease in vastus lateralis cross-sectional area that exceeded the MD (1.40 cm²). The mean decrease in cross-sectional area for the immobilized limb (1.11 cm²) did not exceed the MD.

## Correlation between changes in vastus lateralis echo intensity and cross-sectional area

The correlation between change scores for vastus lateralis echo intensity and cross-sectional area of the immobilized limb was statistically significant ($r = −.649$, $p = .016$ (Fig. 4)).

## Rectus femoris echo intensity

The results from the two-way repeated measures ANOVA indicated that there was no limb × time interaction ($F = .308$, $p = .589$) and no main effects for limb ($F = .046$, $p = .833$) or time ($F = .004$, $p = .953$ (Fig. 2)). Analysis of effect sizes and 95% CIs for mean differences revealed nearly identical echo intensity values for the pretest and posttest for the control

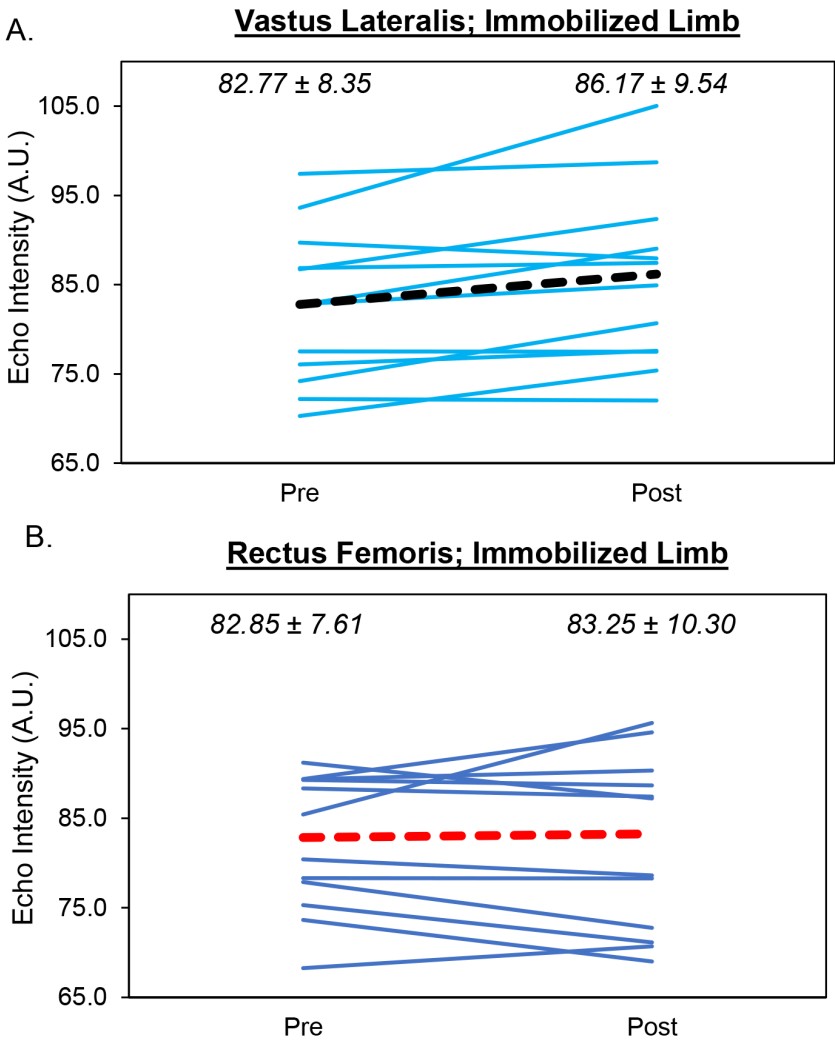

**Figure 2  Changes in Muscle Quality During Limb Immobilization.** Individual participant data showing changes in echo intensity for the vastus lateralis (A) and rectus femoris (B) of the immobilized limb. The thick, dotted lines correspond to the mean values. The mean ± SD values are displayed for each muscle as well.

($d = .087$, 95% CI [$-3.34$–$4.47$ AU]) and immobilized ($d = .084$, 95% CI $= -3.32 -2.51$ AU) limbs. For the immobilized limb, none of the participants showed an increase in rectus femoris echo intensity that exceeded the MD (11.68 AU). The mean increase in echo intensity for the immobilized limb (0.41 AU) did not exceed the MD.

### Rectus femoris cross-sectional area

The results from the two-way repeated measures ANOVA indicated that there was no limb × time interaction ($F = .041$, $p = .794$) and no main effects for limb ($F = .067$, $p = .801$) or time ($F = .450$, $p = .515$ (Fig. 3)). Analysis of effect sizes and 95% CIs for mean differences revealed similar cross-sectional area values for the pretest and posttest for the control ($d = .086$, 95% CI [$-0.41$–$0.55$ cm$^2$]) and immobilized ($d = .309$, 95% CI

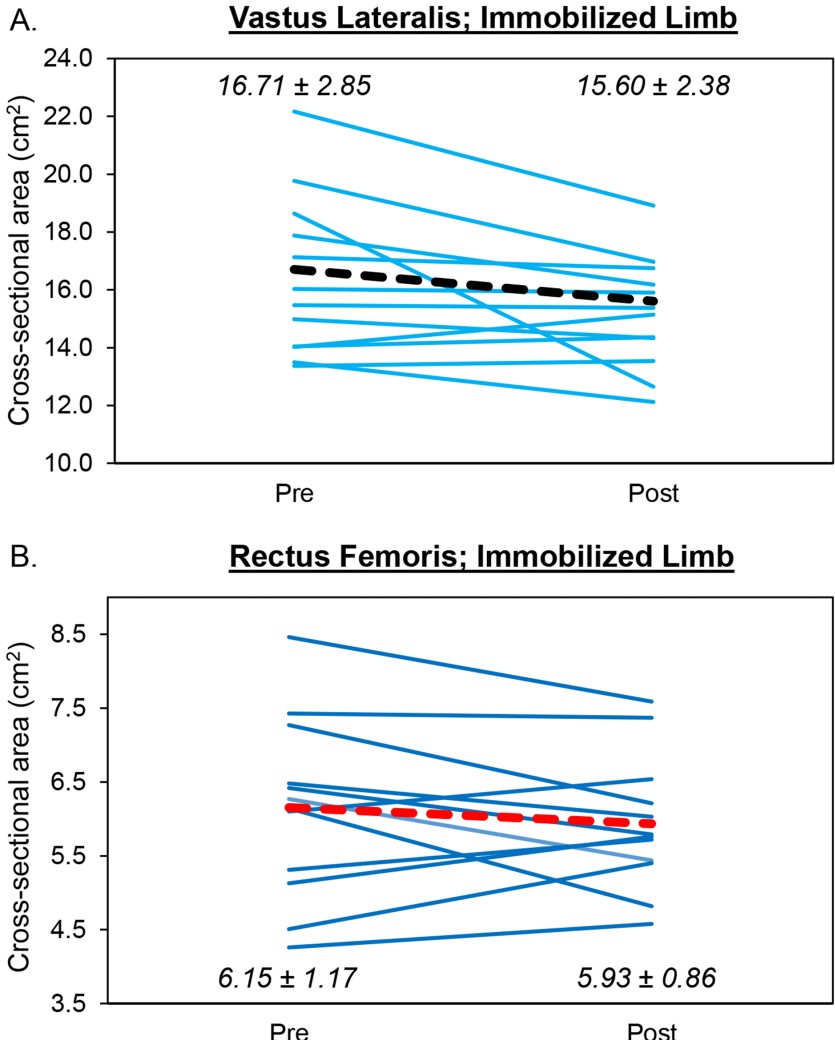

**Figure 3** **Changes in Muscle Size During Limb Immobilization.** Individual participant data showing changes in cross-sectional area for the vastus lateralis (A) and rectus femoris (B) of the immobilized limb. The thick, dotted lines correspond to the mean values. The mean ± SD values are displayed for each muscle as well.

[−0.21–0.65 cm$^2$]) limbs. For the immobilized limb, four participants showed a decrease in rectus femoris cross-sectional area that exceeded the MD (0.67 cm$^2$). The mean decrease in cross-sectional area for the immobilized limb (0.22 cm$^2$) did not exceed the MD.

## Correlation between changes in rectus femoris echo intensity and cross-sectional area

The correlation between the change in rectus femoris echo intensity and cross-sectional area of the immobilized limb was not statistically significant ($r = .378$, $p = .202$ (Fig. 4)).

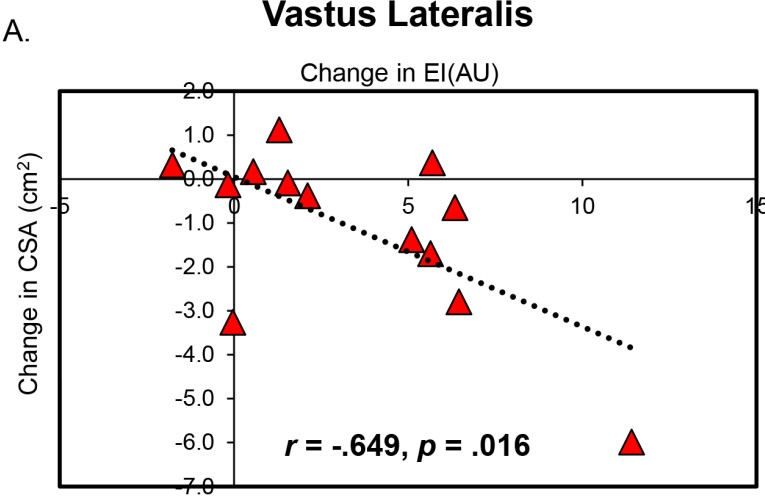

**Figure 4 Significant Associations Between Changes in Muscle Quality and Size.** Scatterplots display the statistically significant correlations between changes in muscle cross-sectional area (CSA) and echo intensity (EI) for the vastus lateralis (A) and rectus femoris (B). It is important for the reader to keep in mind that changes in rectus femoris CSA and EI were small and likely not meaningful.

## Compliance

After removal of one participant because of difficulty with image analysis, the results from the wear-time validation software indicated that all but one participant met the minimum wear-time criteria (8 days of 10+ hours), with participants wearing the device for an average of $13.1 \pm 1.3$ days. Additionally, significant differences existed between minutes of vigorous physical activity (mean difference of 70.2 min $\pm$ 52.8; $p = 0.002$) and total steps (mean difference of 14,861 steps $\pm$ 19,535; $p = 0.04$) between the immobilized limb and the control limb.

## DISCUSSION

Echo intensity is becoming increasingly utilized by investigators to assess changes in skeletal muscle composition (i.e., muscle quality) during various interventions. The major finding of the present study was that two weeks of left knee joint immobilization resulted in a significant increase in echo intensity for the vastus lateralis muscle, and this change was slightly greater than that for the decrease in cross-sectional area. No changes were found for the rectus femoris muscle, which is consistent with previous findings (*Hackney & Ploutz-Snyder, 2012*) and its continued role as a hip flexor throughout the two week intervention. Overall, these results suggest that echo intensity, which appears to reflect intramuscular adiposity (*Reimers et al., 1993*; *Young et al., 2015*) and fibrous tissue infiltration (*Arts et al., 2012*; *Pillen et al., 2009*), may be a more sensitive measure of muscle morphology than cross-sectional area when studying limb disuse. It is important to note, however, that the magnitude of these changes varied across participants, as many of the pretest–posttest differences did not exceed our laboratory's MD statistics.

The results of the present study supported the hypothesis that there would be greater changes in echo intensity compared to cross-sectional area. Although our study design did not include ultrasonography testing at multiple time points, these findings suggest that the deterioration in muscle quality may occur early in the disuse period and –perhaps more importantly—prior to the onset of notable atrophy. To our knowledge, our two week study is the first to find potential indications of adipose tissue changes following as little as two weeks of disuse. To our understanding, only two previous studies have investigated infiltration of intramuscular adipose tissue following a period of disuse (*Gorgey & Dudley, 2007*; *Manini et al., 2007*). *Manini et al. (2007)* found a 14.5% increase in intermuscular adipose tissue in the thigh musculature following a four-week period of unilateral lower limb suspension. Of interesting note, this accumulation of intermuscular fat occurred prior to any significant changes in subcutaneous fat content. The hypothesized physiological explanation for the increased intermuscular fat content was due to a suppression of skeletal muscle lipase activity during periods of inactivity. Similar findings were found in participants who had been diagnosed with an acute, incomplete spinal cord injury (*Gorgey & Dudley, 2007*). The disuse of the lower extremities occurred in a more clinical manner, in which the participants were primarily wheelchair dependent due to the nature of the injury. Substantial increases in intramuscular adipose tissue were found in the thigh musculature six weeks after the injury. The largest and most dramatic increase was seen during the initial six weeks, but there was continued infiltration of intramuscular fat at three months follow up (*Gorgey & Dudley, 2007*). While other studies specifically addressing changes in muscle quality following disuse are limited, insight into this topic may be gleamed from the aging literature, as aging and inactivity seem to manifest in similar physiological adaptations (*Bell et al., 2016*). Specifically, high echo intensity values have been associated with muscle weakness in the elderly population (*Watanabe et al., 2013*). Muscle quality in older adults has also been shown to be a significant predictor of performance during the 30-second chair stand test (*Lopez et al., 2017*; *Rech et al., 2014*), which has important functional relevance. Given that periods of disuse are common in many medical scenarios, such as ACL tears,

our findings highlight the need for researchers and health care professionals to focus their attention on deterioration of muscle quality, rather than just atrophy. A post-operative ACL reconstruction patient, for whom weight bearing is contraindicated, may suffer a reduction in muscle quality despite no noticeable change in muscle size. These results suggest that further research is warranted to investigate early interventions to mitigate the effects of short-term disuse on muscle quality.

The moderate decline in cross-sectional area observed in this study is in line with a variety of previous studies. Following a comprehensive review of unilateral limb immobilization studies, *Hackney & Ploutz-Snyder (2012)* determined that the vastus lateralis was significantly more prone to disuse than the gastrocnemius and the rectus femoris. Based on the literature available at the time, it was estimated that the vastus lateralis atrophies at a rate of .44% per day. Our design and results seem to be quite similar to the work of *Wall et al. (2014)*, who also studied a two-week duration. These authors reported significant atrophy of the vastus lateralis after only five days of immobilization, demonstrating a 3.5% decrease. The decline in cross-sectional area continued as the period of immobilization prolonged, reaching an 8.4% decrease by 14 days. It should be noted that our focus was on healthy females, primarily due the relevance to knee joint injuries in athletics (*Prodromos et al., 2007*; *Renstrom et al., 2008*). Additional studies are needed to investigate if there are differences between sexes and other muscle groups in the ability to preserve muscle quality versus size during a period of disuse. Given that the decline in muscle size was lesser in magnitude than that observed for muscle quality, it is evident that declines in muscle function during periods of immobility are mechanistically related to a variety of physiological phenomena.

To the best of our knowledge, this is the first study to utilize ultrasonography to track changes in echo intensity following limb immobilization. Previous immobilization studies have primarily utilized MRI (*Berg, Larsson & Tesch, 1997*; *Gorgey & Dudley, 2007*) and CT-scans (*Dirks et al., 2014*; *Wall et al., 2014*) to analyze decreases in cross-sectional area of the quadriceps femoris muscles. While our results provide additional support for the use of ultrasonography-based echo intensity as a time- and cost-effective means of studying muscle morphology, we believe that its exact anatomical underpinnings require further investigation. On one hand, there is strong evidence to support the notion that echo intensity is associated with intramuscular adiposity (*Arts et al., 2012*; *Young et al., 2015*). A study by *Young et al. (2015)* conducted MRI and ultrasonography imaging on 31 participants of varying ages for the rectus femoris, biceps femoris, tibialis anterior, and medial gastrocnemius. They found that there were strong correlations between the percentage of intramuscular fat as shown with MRI and echo intensity as measured with ultrasonography, with the rectus femoris association being particularly high ($r = .91$). On the other hand, the influence of fibrous tissue content should not be completely discounted. *Pillen et al. (2009)* analyzed both echo intensity and muscle biopsies of golden retrievers suffering from muscular dystrophy. The results of their study revealed that muscle echo intensity was strongly correlated ($r = .87$) with the amount of intramuscular fibrous tissue. In addition, the fact that echo intensity is altered by muscle tissue damage following eccentric exercise (*Nosaka & Sakamoto, 2001*) and potentially reflective of

glycogen/hydration status (*Jenkins, 2016*) points to the need for more comprehensive studies in this area. Collectively, while the term muscle quality is likely an appropriate term for ultrasonography-based echo intensity, the precise explanation for the changes observed in the present study is worthy of additional inquiry.

Like all studies, this investigation had several limitations that are worthy of brief discussion. First, although our efforts to monitor and ensure compliance give us confidence in these findings, it seems impossible to state with 100% certainty that our participants did not ever weight bear. Future investigators wishing to perform similar studies in college-aged participants should consider the use of a golf cart to provide transportation to and from classes. Unfortunately, this option was not available when the present study began. Second, while we were motivated to perform this study in young females, the external validity to other populations is somewhat limited. Indeed, future studies are needed in young males as well as varying age groups to determine if the enhanced sensitivity of echo intensity versus cross-sectional area is consistent across sexes and age groups. As echo intensity appears to be lower in African Americans (*Melvin et al., 2014*), racial and ethnic backgrounds of research participants should also be studied. Third, as our study was designed to specifically examine limb immobilization, these findings should not be extrapolated to other non-weight bearing conditions, and in particular, bedrest. We believe that the ambulatory activities of the control limb should not be discounted, particularly given the influence of cross education training (*Carr et al., 2019*) and the central nervous system's role in maintaining muscle strength and mass (*Clark et al., 2014*). Similarly, our implementation of a 90° joint angle at the knee to ensure that the participants' toes did not come into contact with the ground during ambulating with crutches resulted in a unique biomechanical condition, particularly for the biarticular muscles of the thigh. Finally, it is interesting to note that although some group mean differences and moderate/large effect sizes were demonstrated, only a small portion of the participants showed changes that exceeded the MD. None of the mean differences exceeded the MD. This contrast was despite good-to-excellent test-retest reliability statistics of well-trained and experienced investigators. These results highlight the need to consider change scores on a participant-by-participant basis, as well as the practical value of including multiple and comprehensive analytical approaches in research (i.e., effect sizes, confidence intervals, *p* values, individual participant data, etc.).

## CONCLUSIONS

We sought to examine the effects of two weeks of knee joint immobilization on vastus lateralis and rectus femoris muscle quality and size. The findings of this study demonstrated that muscle quality deteriorated to a greater extent than muscle size during two weeks of knee joint immobilization in college-aged females. We believe that the lack of change for the rectus femoris due to its role as a hip flexor provides further justification for this conclusion. Future investigators may wish to verify these results in males and participants of various age and ethnic groups. Based on these data, ultrasonography-based echo intensity appears to be a useful clinical tool for tracking changes in skeletal muscle deterioration during extended periods of disuse.

### Funding

Funding support for this study was provided by the De Luca Foundation Research Scholarship program to Mr. Rob MacLennan, as well as the University of Central Florida's Advancement of Early Career Researchers program to Dr. Matt Stock. The funders had no role in study design, data collection and analysis, decision to publish, or preparation of the manuscript.

### Grant Disclosures

The following grant information was disclosed by the authors:
De Luca Foundation Research Scholarship program.
University of Central Florida's Advancement of Early Career Researchers program.

### Competing Interests

The authors declare there are no competing interests.

### Author Contributions

- Rob J. MacLennan and Matt S. Stock analyzed the data, conceived and designed the experiments, performed the experiments, prepared figures and/or tables, authored or reviewed drafts of the paper, and approved the final draft.
- Michael Sahebi, Nathan Becker and Ethan Davis analyzed the data, conceived and designed the experiments, performed the experiments, feedback and edits of manuscript, and approved the final draft.
- Jeanette M. Garcia analyzed the data, performed the experiments, feedback and edits of manuscript, and approved the final draft.

### Human Ethics

The following information was supplied relating to ethical approvals (i.e., approving body and any reference numbers): The University of Central Florida's Institutional Review Board approved all aspects of this study (Study # BIO-17-13642).

### Data Availability

The following information was supplied regarding data availability: All of the ultrasound data for this study is available in a Supplemental File.

### Supplemental Information

Supplemental information for this article can be found online at http://dx.doi.org/10.7717/peerj.8224#supplemental-information.

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
