# Peer review of "Declines in skeletal muscle quality vs. size following two weeks of knee joint immobilization"

_PeerJ, doi:10.7717/peerj.8224_

## Round 0.1 · original submission · Major Revisions

The reviewers generally commented positively on the manuscript and provided constructive feedback that should be used to further strengthen the manuscript. In particular, the reviewers raised some questions about the depth of the signal with ultrasound, the length and type of immobilization strategies and the reported statistical, which should be clarified in a revised version.

Reviewer 1 ·

Basic reporting

No comment.

Experimental design

No Comment

Validity of the findings

No comment

Additional comments

Muscle quality was examined by isolating the rectus femoris and vastus lateralis to intentionally cause atrophy within the muscle and examine the cross sectional area and echo intensity within the brace. The study was completed in college-aged females because they are more prone to ACL and MCL tears throughout sports. It was found that a decline in muscle size was lesser in magnitude than that observed for muscle quality. I appreciate the detail of methods as they could easily be replicated.
1. The introduction provides a concise overview of the muscular changes to immobilization, the use of muscle quality, and the purpose of this study, however, it does not provide a real justification of why understanding these morphologies are important or useful. For example, if echo intensity is reported in arbitrary units, to what extent are the values important to a practitioner, unless it is purely within that independent clinic (and not compared between independent cohorts or clinicians that may use different settings to record the data.) Can the authors please elaborate on the significance of this type of measurement and perhaps how it can be generalizable to all?
2. It is my understanding that altering the depth of the signal with ultrasound will then alter the gray scale of the image. The authors report a scan depth of 4-7 cm, individualized for each participant. If the scan depth were to remain constant (6 cm-for example) and then normalized for subcutaneous fat, then the results will be more consistent when evaluating echo intensity.
3. Was menstrual phase recorded? The hormonal shifts estrogen during the follicular (higher) and luteal (lower) phases may alter the muscle-sparing response that naturally occurs in women.
Enns DL, and Tiidus PM. The influence of estrogen on skeletal muscle. Sports Medicine. 2010.
Wikström-Frisén L, Boraxbekk CJ, Henriksson-Larsén K. Effects on power, strength and lean body mass of menstrual/oral contraceptive cycle based resistance training. Journal of Sports Medicine and Physical Fitness. 2015.
4.The RF and the VL (although different in size) are both biarticular muscles. Would the flexed leg at 90 degrees and perhaps participants flexing at the hip impact the use of these muscles? RF would potentially be stimulated nearly as much with the anterior aspect of the thigh and proximity to the psoas muscles. This could be a potential reason for lack of noticeable change in the RF.
5. The tables and charts are redundant with the text. I think the table can be translated onto the line charts and then be eliminated. Also, the mean lines of the charts should indicate standard error bars/margins.

Reviewer 2 ·

Basic reporting

All basic information meets PeerJ standards

Experimental design

The experimental design is fully adequate to answer the research question. Attrition rate and exclusion in results is clearly explained. Methods are meticulous and clear.

Validity of the findings

It appears that decline in echo intensity is a individual basis.Thee participants showed a decline but majority did not show a change in EI greater then the MD. Previous studies showed that unweighted legs showed increase in adopisity. Do to the nature and difference in immobilization strategies. Both are logical ways to unload the limb but may have many different neural components. Brace and position may actually influence more neural activity. Authors need to explain more why 90 degrees was chosen and possibly the influence this has with a elongated MTU.

Additional comments

Over all finding that immobilization has a greater affect on quality then size does contribute to the literature. Need to clear up the discussion as to differences observed and fi duration of 2 weeks is long enough to see adipose changes. Further explanation on why 90 degrees was chosen needs to be addressed.

·

Basic reporting

Regarding the five PeerJ points, i.e., language, context and references, structure, figures, and raw data, the current manuscript meets all. I just have some minor suggestions in their context further presented in general comments of this review.

Experimental design

Among the four PeerJ points about experimental design, the current manuscript presented an original line of investigation and a well-defined research question. In addition, human research ethical was respected. Regarding the methods, further suggestions were provided in general comments of this review.

Validity of the findings

PeerJ listed four points for the validity of findings. The interpretations of findings and some decisions will be critically appraised in general comments of this review. In short, there is some comments and suggestions on results and some speculations, but all aiming to help reach a better level of publication.

Additional comments

Please, find in the attachment my general comments.

---

## Round 0.2 · Minor Revisions

While the revised manuscript has improved considerably, the reviewers have raised some additional minor comments that need to be addressed before the manuscript can be accepted for publication.

Reviewer 1 ·

Basic reporting

no comment

Experimental design

no comment

Validity of the findings

no comment

Additional comments

I commend the authors for their revisions. The manuscript clarity and translation to broad impacts in support of future research with use of ultrasound imaging. I have one remaining concern/comment.
-Although the authors provide support for the correction of RF values, the general change between the RF and VL should mirror one-another. I feel the corrected and UNCORRECTED RF values should be represented in the manuscript so a more direct comparison of the two muscles can be drawn.
-Additionally, I'm curious how many participant's image depth deviated from 4cm? If they were removed fron analyses, would anything change? Thank you for providing the raw data in the response to reviewers.

Reviewer 2 ·

Basic reporting

Meets basic Reporting

Experimental design

Questions regarding methods have been addressed with the changes.

Validity of the findings

Authors address the validity concerns of exceeding MD values and have clarified reliability statistics. With a novel approach and slightly different immobility set up as with all human research is not going to be perfect. The finding do start to shed light on the risks and physiological costs of immobilization.

Additional comments

Authors have rewritten the discussion to clarify the duration of disuse and possible implications.

·

Basic reporting

Meets all basic reporting.

Experimental design

Meets all experimental design criteria.

Validity of the findings

Meets all validity of the findings.

Additional comments

General comments
The manuscript “Declines in skeletal muscle quality vs. size following two weeks of knee joint immobilization” was improved after the last revision round. The authors should be commended for their efforts. All my comments were properly answered, and I just have a few minor suggestions that may help to improve Methods section:
1) Sample size – I agree to the authors regarding the incapability to estimate effect size for muscle echo due to the lack of previous data. Considering the aforementioned, it would be great for future studies if the authors provided the power reached with their sample size.
2) Ultrasonography Measurements and Analysis – Great. The authors have interesting data about the variability of muscle EI across different depths. To ensure readers the full information, I would like to suggest the authors to state briefly the results of this subsample, describing the coefficient of variation, and standard error mean. In addition, it would really great if the authors also indicate how many participants the depth had to be changed. It is really great that you have such important information for future trials and investigations.
I do not have any more comments/suggestions, and I would like to suggest the acceptance of this manuscript with the consideration of previous suggestions. Congratulations to the authors.

---

## Round 0.3 · accepted · Accept

The authors have adequately addressed the remaining conments

Reviewer 1 ·

Basic reporting

No Comment

Experimental design

No Comment

Validity of the findings

No Comment

Additional comments

Thank you for taking the time to make the suggested edits. Well done and congratulations.